# Exosomes; a Potential Source of Biomarkers, Therapy, and Cure for Type-1 Diabetes

**DOI:** 10.3390/ijms242115713

**Published:** 2023-10-28

**Authors:** Jonathan R. T. Lakey, Yanmin Wang, Michael Alexander, Mike K. S. Chan, Michelle B. F. Wong, Krista Casazza, Ian Jenkins

**Affiliations:** 1Department of Surgery, University of California Irvine, Irvine, CA 92617, USA; michaela@hs.uci.edu; 2Department of Biomedical Engineering, University of California Irvine, Irvine, CA 92617, USA; 3California Medical Innovations Institute, 11107 Roselle Street, San Diego, CA 92121, USA; yanmin-wang@calmi2.com; 4Uropean Wellness Group, Klosterstrasse 205ID, 67480 Edenkoben, Germany; michael.chan@klgates.com (M.K.S.C.); michelle.wong@klgates.com (M.B.F.W.); 5Baden R&D Laboratories GmbH, z Hd.v. Sabine Conrad, Ferdinand-Lassalle-Strasse 40, 72770 Reutlingen, Germany; 6GATC Health Inc., Suite 600, 2030 Main Street, Irvine, CA 92718, USA; krista.casazza@gmail.com (K.C.); ian@gatchealth.com (I.J.)

**Keywords:** diabetes, stem cells, exosomes, T1D, extracellular vesicles, miRNA

## Abstract

The scourge of type-1 diabetes (T1D) is the morbidity and mortality it and its complications cause at a younger age. This propels the constant search for better diagnostic, treatment, and management strategies, with the ultimate quest being a cure for T1D. Recently, the therapeutic potential of exosomes has generated a lot of interest. Among the characteristics of exosomes of particular interest are (a) their regenerative capacity, which depends on their “origin”, and (b) their “content”, which determines the cell communication and crosstalk they influence. Other functional capacities, including paracrine and endocrine homeostatic regulation, pathogenic response ability resulting in insulin secretory defects or β-cell death under normal metabolic conditions, immunomodulation, and promotion of regeneration, have also garnered significant interest. Exosome “specificity” makes them suitable as biomarkers or predictors, and their “mobility” and “content” lend credence to drug delivery and therapeutic suitability. This review aims to highlight the functional capacities of exosomes and their established as well as novel contributions at various pathways in the onset and progression of T1D. The pathogenesis of T1D involves a complex crosstalk between insulin-secreting pancreatic β-cells and immune cells, which is partially mediated by exosomes. We also examine the potential implications for type 2 diabetes (T2D), as the link in T2D has guided T1D exploration. The collective landscape presented is expected to help identify how a deeper understanding of exosomes (and their cargo) can provide a framework for actionable solutions to prevent, halt, or change the very course of T1D and its complications.

## 1. Introduction

The term “Exosomes” was coined by Johnstone et al. [1]. Exosomes were initially seen as garbage released by cells until 1996, when they were identified as contributors to cell-to-cell communication [1,2]. Cell communication is fundamental to achieving perception and response to stimuli (mechanical/chemical/hormonal/neuronal), maintaining cell/tissue homeostasis, regeneration, repair, and development, and immunity [3]. Prior to this observation, hormones, immune cells, and neurons were considered the sole key players in cell-to-cell communication. However, developments in nanotechnology helped isolate and characterize exosome as the fourth key player [2]. Following the observation that Exosomes participates in cell-to-cell communication, an antigen-presenting role was also identified (Raposo et al., 1996 [4]), suggesting a potential role in autoimmune regulation related to errors in communication. Errors in communication play a key role in derangement in cell/tissue homeostasis and the pathogenesis of a wide range of diseases, including type-1 diabetes (T1D) as well as T1D complications [5,6]. After their release from cells, exosomes transfer lipid, protein, and nucleic acid cargo from cells, mediating interaction with receptors available on the surface of target cells. As such, it is highly plausible that exosomes play an integral role in affecting glucose and lipid metabolism in T1D, insulin delivery, β-cell function, and gene delivery. The prevalence of morbidity and mortality due to the development of T1D complications (hypoglycemia, ketoacidosis, microvascular, and macrovascular complications) are highly related to the severity of impairments and age at onset (Table 1, Figure 1) [7,8,9,10,11,12,13,14]. Interdisciplinary exosome research has demonstrated the importance of exosomes across disease states. For example, exosomes from prostate cancer patients undergoing radiotherapy, chronic atrophic gastritis, vector-borne diseases, and others have emerged as candidate biomarkers for monitoring treatment response and chronic atrophic gastritis. Thus, exosome production, biogenesis, and release routes represent a variety of roles in cellular processes and have the capacity to act as therapeutic agents, promoting a variety of functions and employing indicators or therapy options for several diseases, far from the perceived “junk.”

Despite significant emerging research understanding the mechanistic underpinnings of exosomal biogenesis, there are still unanswered questions related to the role of exosomes in T1D. Exosomes helped answer burning questions at the immuno-molecular and cellular levels. How were nucleic acids and proteins from parent cells able to have a paracrine and even an endocrine effect on target cells? How were they able to remain stable and escape the DNase, RNase, and proteinases present in body fluids or media? The mobility of the exosomes and their clathrin-coated lipid-bilayer cum protein membrane provided the answer [5,6]. Therefore, could a better understanding of exosomes and the crosstalk of their cargo represent utility for prevention, halting, or changing the course of T1D and its complications via early diagnosis, effective treatment, and management, thereby improving outcomes? For example, exosomes promote both health and disease depending on their content, which in turn varies with the cell type and environmental conditions. Through interactions of the vesicles themselves or their cargo, could content be engineered, the biogenesis of exosomes possessing a desired attribute be upregulated, or RNA interference (RNAi)-based therapy using the exosomes as vehicles be used to alleviate, trigger, attenuate, diagnose, or treat? More importantly, where are we in terms of the physiological link between exosomes and T1D? This review will attempt to collate, be concise, and present studies that have attempted to link exosomes and T1D and their potential for therapeutic use.

The review encompasses recent literature using a comprehensive electronic literature search. The search engines used were Google, Google Scholar, Medline/Pubmed, Exocarta. The criteria used to select journal publications for the present study were that the studies should target T1D and clearly mention the term exosome(s) or include the term extracellular vessicles (EVs) in the size range 40–100 nm, and that exosomes were characterized using standard markers. Articles on exosome(s) and other specific types (at times referred to as Type-3 A to H) of diabetes/pancreatic cancers/whole pancreas/islet transplantation/stem cell therapy were not included in the present study [3]. Original articles (experimental, clinical trials) and reviews that matched the key words, were in the English language, were available in full text, and were free were included in the review as relevant. We translated one review article that was in Chinese using Google Translate as it was pertinent to the current topic [15]. To ensure a comprehensive review of relevant literature, key words used included exosome(s) and β-cells, exosomes and islets, and exosome(s) and T1D. The particulars of the free articles available in full text based on the keywords used in our search were as follows: exosome + β-cells: Full texts: 44 (Free: 32); Clinical Trial; Review: 18 (Free: 15); exosome + Islets: Full texts: 55 (Free: 34); Clinical Trial:-; Review: 12 (Free:6); exosome+ T1D: Full texts: 54 (Free: 37); Clinical Trial:-; Review: 16 (Free: 10). In certain instances, articles that were relevant to the current paper yet covered T1D and T2D, or work on islets and stem cells, were included in our study. Pertinent articles on exosome biogenesis and function, the current epidemiology of T1D and T2D, and their complications were also included as appropriate.

Rapid Glance at the Statistics on Type 1 Diabetes (T1D) [7,8,9,10,11,12,13,14].

Impairment/Hallmark: Insulitis, Autoimmune Response, Progressive β-cell destruction resulting in frank hyperglycemia, Cytokine network.Worldwide estimates of annual incidence of T1D was 128,900 new cases (<20-year age-groups) with corresponding prevalence of 1,110,100 for a Total Population (1000s) of 2,582,088.Proportion of youth in the T1D Exchange Clinic Registry in 2013 failing to meet the American Diabetes Association (ADA) and International Society for Pediatric and Adolescent Diabetes (ISPAD) targets were: HbA1c (79%), Systolic and diastolic blood pressure (SBP/DBP) (22%), LDL-cholesterol (LDL-C) (38%), Triglycerides (TG) (11%) and Body mass index (BMI) (31%)The Diabetes Control and Complications Trial (DCCT) which randomized people with T1D to receive intensive or standard insulin therapy; after 21 months 216 of 817 subjects had already experienced a severe hypoglycemic episode.Prevalence of Diabetic Retinopathy (DR) in the Wisconsin Epidemiologic Study of DR (WESDR) was 17% (<5 years duration of T1D) and 98% in those with (≥15 years duration of T1D)Elevated albumin excretion 50% in T1DPrevalence of hypertension in youth with T1D: 4–7% compared to those without T1D: 1–5%In the SEARCH for Diabetes in Youth Research Group prevalence of diabetic kidney disease was 6%, arterial stiffness 11% and hypertension 10% in youth with T1D (51). Prevalence of overweight in T1D (age 3–19) was 22.1% compared to their nondiabetic peers 16.1%)

## 2. Exosomes

Exosomes are 30–200 nm extracellular vessicles (EVs), density: 1.13–1.19 g/mL, secreted by all cell types, with a clathrin-coated lipid-bilayer cum protein membrane, impermeable to propidium iodide, formed via endocytosis [16,17,18,19,20]. Exosomes encapsulate a diverse array of biomolecules, including proteins, lipids, nucleic acids (such as microRNAs and messenger RNAs), and metabolites. The cargo encapsulated within the exosome is selectively sorted, reflecting the cell of origin’s physiological state. Currently, bone marrow-derived mesenchymal stem cells (MSCs) and placenta tissue-derived cells are the primary cell sources of exosomes for commercial products. In comparison, bone marrow-derived MSCs have a much higher safety profile and yield higher-quality exosomes that deliver a more favorable profile of signaling factors [21]. Exosomal composition and concentration depend on (a) their parent cell type, (b) the microenvironment (normal, stressful (hypoxic/ER stress/hyperglycemic/ischemic), and (c) tumorigenic versus non-tumorigenic versus normal status [16,17,21,22,23,24,25,26,27,28,29]. The unique characteristics of exosomes, including their ability to encapsulate diverse cargo and traverse biological barriers, have sparked considerable interest in their clinical applications. Exosomes hold promise as non-invasive biomarkers for disease diagnosis, prognosis, and monitoring due to their stability in bodily fluids. Additionally, exosomes are being explored as vehicles for targeted drug delivery, as they can be engineered to carry therapeutic payloads and exhibit a propensity for homing to specific cell types.

Standard methods to isolate and purify/concentrate exosomes include ultracentrifugation (100,000–200,000× *g*), filtration, density gradient immunoprecipitation size exclusion chromatography, and commercial kits [30,31,32]. While quantification and size estimation of exosomes are done via dynamic light scattering, nanoparticle tracking analysis, and surface plasmon resonance [30,31,32], these methods represent acceptable methodological approaches in the growing interest in identifying exosomal biomarkers. Among the membrane and transmembrane proteins, major histocompatibility complex (MHC) molecules, tetraspanins, adhesion molecules, and metalloproteinases are exosome sub-populations. Exosomes are the subpopulations of EV that test CD9-, CD63-, and CD81-positive (tetraspanins) [20,21,22]. Exosomal markers include those associated with membrane transport and fusion (Rab GTPases, Annexins), tetraspanins (CD63, CD9, and CD81), and MVB-biogenesis-associated proteins (Alix and TSG101) [33].

## 3. Exosome Biogenesis, Secretion, and Communication/Interaction with Target Cell/s

Briefly, the early endosome (Figure 2) is formed by inward budding of the plasma membrane. It in turn pinches inward, forming many intraluminal vesicles (ILVs) to transform into a multivesicular body (MVB) [34]. The highly specific cargo sorting within the ILVs is driven either by endosomal sorting complexes required for transport (ESCRT: 0, I, II, and III) or by the ESCRT-independent ceramide-requiring pathway [5,34,35,36,37,38,39,40]. If the MVB fuses with a lysosome, it is digested. However, if energy-dependent fusion with the plasma membrane with the aid of Rab GTPases and SNARE proteins occurs, releasing the ILVs into the extracellular medium, these ILVs are termed exosomes [5,36,41,42,43,44,45] (Figure 2). Circulating endosomes have a half-life ranging from 2–5 min to three hours in certain cases [33,46,47,48].

Exosomes influence cell-to-cell communication locally and systematically, thus exhibiting both a paracrine and endocrine effect as they are carried by the environment into which they are secreted (media, urine, blood, milk, plasma, lymph, and cerebrospinal fluid) [24,25,49]. This enables them to act as protective shuttles, carrying their precious cargo to the target cells, which include growth factor receptors, transcription factors, regulatory proteins, and functional RNA [5,42,49,50,51,52]. This cargo is protected from DNAses and RNAses by the exosomal membrane, which in turn influence/change the behavior of the target cell, either promoting health or activating pathogenesis following fusion [5,29,33,43,52,53,54,55,56,57,58,59,60,61]. Exosomes are also capable of modifying target-cell phenotypic features [62,63] via the genetic material they contain following fusion with the target cell via multiple mechanisms (Figure 2), i.e., receptor-ligand interactions/direct fusion/endocytosis [64].

(i)The exosome directly fused with the plasma membrane of the target cell, releasing its contents into the cell [65].(ii)Ligand (exosomal membrane)—receptor (target cell membrane) binding triggers a cascading downstream effect [36,66,67,68,69].(iii)Receptor-mediated endocytosis of the exosome or fusion with the plasma membrane [33,36,66,70].

Horizontal transfer of genetic material is brought about following fusion with endosomes to affect entry into the cytosol of target cells. Once entry is gained into the target cell, phenotypic and behavioral changes are affected by (i) direct stimulation via surface-bound ligands, (ii) the transfer of activated receptors, and (iii) epigenetic reprogramming following the transfer of functional proteins, lipids, and RNAs to the target cell [71].

## 4. Exosomes Represent a Potential Missing Link in T1D Therapy

T1D is a complex disorder emanating from the interplay of a myriad of variables, both endogenous and exogenous. Indeed, the use of genome-wide association studies has begun to enhance our understanding of the potential genetic basis of T1D, with ~60 risk regions identified. Recent estimation suggests that approximately half of the risk factors for T1D are genetic (Cerolsaletti, 2019 [72]), extending beyond the notion that susceptibility to developing T1D is influenced by carrying specific high-risk haplotypes of class II human leukocyte antigens (HLA). Notwithstanding, 50% of the risk therefore includes an intricate interaction and homeostatic regulation integrating environmental factors, collectively inducing β-cell stress as the mechanistic underpinning. It is well established that the mechanisms of T1D are initiated significantly prior to clinical manifestation. The prevailing dogma suggests T1D is a complete loss of functional β-cells; however, some T1D patients can produce insulin; nevertheless, the functional β-cell mass is still decreased to the level of insulin dependency. As is well-reported in type 2 diabetes, T1D also may have a “pre-T1D” phase, yet as the immune-mediated destruction of β-cells amasses, the autoimmunity associated with T1D initiates an uncompensated chronic inflammation, further promoting disease progression, β-cell apoptosis, and ultimately T1D. The past century has been met with significant treatment and management options for T1D.

Recently, transplantation of the pancreas or islet β-cells to prevent or slow complications and progression has evolved. As an alternative to pancreas transplantation, islet β-cell transplantation, in which donor islets are infused into the liver via the portal vein, has also emerged as a therapeutic strategy. Islet transplantation does not replace the entire β-cell mass; it is less invasive than a pancreas transplant while still providing the potential to restore normal glycemic function, re-establish hypoglycemia awareness, and minimize severe hypoglycemic responses. As these advances continue to emerge, there are several pieces of evidence that indicate that exosomes have utility in the preservation and survival of pancreatic islet cells prior to complete destruction.

## 5. Exosomes in Autoimmune Type-1 Diabetes (T1D)

Ironically, positivity to glutamic acid decarboxylase (GAD65), tyrosine phosphatase-like protein (IA-2), proinsulin/insulin (IAA), and zinc transporter 8 (ZnT8) (autoantibodies) can signal a risk for T1D, yet many testing positive neither develop insulitis nor T1D [73,74,75]. T1D develops only when balance and control within the innate and acquired immune systems are destroyed following exposure to exogenous triggers (environment, diet, viruses, infections, toxins) and/or epigenetic changes induced [73,74,75]. In a recent study, Horiguchi et al. reported exosome degeneration in T1D adipose-derived stem cells [76].

Due to molecular mimicry between β-cell auto-antigens and viral antigens, on exposure, autoreactive T-lymphocytes are activated, unleashing a cascade of events culminating in insulitis. Briefly (Figure 3), during early insulitis, local antigen-presenting cells (APC) are activated, which recruit CD4+ helper T-cells from the pancreatic lymph nodes, which release chemokines/cytokines [73,74,75]. CD4+ helper T-cells induce APCs to secrete cytokines and nitric oxide and to induce the endothelial cells to secrete chemokines, activating CD8+ cytotoxic T-cells (Figure 3) [73,74,75]. β-cells secreting chemokines in response to a viral infection or cytokines further stimulate and activate immune cells, resulting in Fas pathway and granzyme/perforin system (mitochondrial pathway) activation, which induces β-cell apoptosis. In addition, the cytokines interleukin-1β (IL-1β), tumor necrosis factor α (TNFα), and interferon γ (IFN-γ) directly bind to their respective β-cell surface receptors, and apoptosis is induced (Figure 3) [73,74,75].

Among several theories of how exosomes participate in immunomodulatory and/or communication pathways, exosomes from MIN6 cells and islet-associated MSC–like cells activate T cells and B cells in prediabetic non-obese diabetic (NOD) mice [59,77,78,79,80,81]. Autoantigen presentation to activate T lymphocytes via exosomes is either (a) indirect: first via APCs activating naïve T cells followed by APC co-stimulatory molecules activating T lymphocytes, (b) the exosome is engulfed by APC, following which its complex peptide/MHC is exposed, or (c) direct activation [82].

Autoantigen GAD65, IA-2, and proinsulin/insulin containing primary human and rat islet-derived exosomes activated APCs with the T1D DRB1*0401 haplotype. In parallel, Th1 cytokines IL-1β and IFN-γ secreted by inflammatory cells induced ER stress in β-cells, which in turn released the proinflammatory ER chaperone proteins calreticulin, Gp96, and ORP150 that promote phagocytosis [59,77,78,79,80,81]. ER stress-inducing cytokine treatment of mouse islets increased proinflammatory ER chaperone proteins in INS-1E cells [59,77,78,79,80,81]. Exosomes from T- lymphocytes containing microRNAs (miR-142-3p, miR-142-5p, and miR-155) when transferred to rodent and human pancreatic β cells triggered genes coding for chemokines (Ccl2, Ccl7, and Cxcl10) in pancreatic β-cells, promoting apoptosis and defects in insulin secretion following activation of nuclear factor κB (NF-κB) [82]. The expression patterns and functions of microRNA were similar between human umbilical cord blood-derived MSC and MSC-derived exosomes [83]. In addition to microRNAs, plasma-derived exosomal circular RNA, long non-coding RNA, and messenger RNA all play roles in T1D pathogenesis and need further exploration [84,85,86].

## 6. Exosomes in Health, and T1D Prevention as Therapeutics

Exosomes promote cell/tissue health by regulating oxidative stress, apoptosis, proliferation, angiogenesis, inflammatory responses, and tissue repair [87,88]. One example of the dualistic property of exosomes is that low cytokine concentrations stimulated neutral ceramidase (NCDase) release, preventing the **concentration-dependent effect of inflammatory cytokines** apoptosis via the NCDase-S1P-phosphate-S1P receptor 2-dependent mechanism. At high cytokine concentrations, NCDase release was inhibited in rat β-cell line INS-1 cells, and apoptosis occurred [89].

An example of the dualistic **immunomodulatory property of the exosome** is that MSCs-derived exosomes from healthy donors (from bone marrow) suppress TNF-α and IL-1β (pro-inflammatory factors) but increase TGF-β (anti-inflammatory factors) in vitro [15,74,80,90]. MSCs-derived exosomes also induce T-helper type-1 (Th1) cells to convert to T-helper type-2 (Th2) cells, thus preventing their conversion into interleukin-17-producing effecter T-cells (Th17) [15,74,80,90]. Exosomal MSCs also inhibit autoimmunity, preventing T1D by inhibiting Th1 and Th17 cells but increasing the expression of IL-10, an immunosuppressive cytokine [15,74,80,90]. Exosomes derived from adipose tissue-derived MSCs have shown ameliorative effects on T1D by increasing the population of regulatory T-cells and their products without altering the proliferation index of lymphocytes [91]. This makes them more effective and practical candidates. In terms of the potential of exosomes in T1D treatment, exosomes from β-cell lines increased survival time at high doses (H-STZ, 200 mg/kg) STZ-mice alleviated glucose intolerance in low doses (L-STZ, 140 mg/kg). STZ-treated C57BL/6J mice showed increased CD31 expression and decreased macrophage infiltration, indicative of their neovascularization and immuno-modulatory ability [92].

The **crosstalk between islets and endothelial cells was studied bi-directionally**. Human islet exosomes containing angio miRs (miR-27b, miR-126, miR-130, and miR-296) and mRNAs of genes associated with angiogenesis (eNOS and VEGFa) when incubated with islet endothelial cells (IEC) induced the expression of pro-angiogenic genes (angiopoietin1, VEGFa, VEGFR1, and VEGFR2) while repressing anti-angiogenic thrombospondin1. They prevented apoptosis by promoting anti-apoptotic factors (BCL-2) while repressing pro-apoptotic Bcl-2 agonists of cell death (BAD) [93]. The supernatant of human endothelial progenitor cells (EPC) promotes insulin release and β-cell survival, induces IEC proliferation and migration, and facilitates the formation of capillary-like structures called islet grafts [94]. Sharma et al. [83] discovered that in T1D mice, the MSC-derived exosomes attenuated hyperglycemia and pancreatic injury while promoting the proliferation of islet cells and insulin production. However, the relevant pathways were different from those observed in MSC therapy. Sabry et al. [95] treated T1D rats with either MSCs or MSC-derived exosomes. To assess the regeneration of pancreatic beta cells, insulin, pancreatic and duodenal homeobox 1, Smad 2, Smad 3, and transforming growth factor beta (TGFβ) genes were measured. Histopathological and immune-histochemical examinations were performed to confirm pancreatic tissue regeneration. The results indicated that MSC-derived exosomes demonstrated superior therapeutic and regenerative effects compared to MSCs alone [95].

Among studies on the **biomarker potential of exosomes**, one study by Lakhter et al. [96] evaluated the biomarker potential of miR-21-5p, found to be elevated and resulting in increased β-cell apoptosis in rodent and human models in vitro, in vivo, and via a clinic study [59,96,97,98,99,100,101]. For the in vitro studies, β-cell lines (MIN6 and EndoC-βH1) and human islets were exposed to cytokines (IL-1β, IFN-γ, and TNF-α) to mimic autoimmune T1D. The supernatant was assayed for miR-21-5p levels and showed a 1.5 to 3-fold increase in miR-21-5p levels with a 3 to 6-fold increase in EV miR-21-5p. NOD mice that spontaneously develop T1D were used for in vivo studies from 8-weeks until the onset of diabetes, and serum was collected on a weekly basis [96]. A positive correlation was found between EV miR-21-5p levels and the onset of diabetes in NOD mice, especially three weeks prior to the onset of diabetes [96]. For the clinic study, sera from 19 children with T1D and 16 healthy controls were evaluated, and levels of miR-21-5p from serum EVs were found to be 3-fold higher than in nondiabetic individuals, while total serum miR-21-5p was lower in subjects with T1D [96]. Long non-coding RNAs (lncRNAs) involved in transcriptional, epigenetic, and post-transcriptional gene regulation functions were upregulated in pre-diabetic NOD mouse islets and in exosomes of MIN6 cells following pro-inflammatory cytokine pretreatment. Exosomes have also been identified as biomarkers of diabetic complications in T1D. Kalani et al. [102] reported that urinary exosomal Wilm’s tumor-1 protein in T1D patients could serve as an early non-invasive marker for diabetic nephropathy. In addition, C5b-9 levels in serum astrocyte-derived exosomes may be used as a marker of cognitive impairment in T1D patients [103].

Among studies on **complications due to T1D**, exosomes from human urine-derived stem cells **prevented T1D nephropathy** via the transfer of transforming growth factor-β1, angiogenin, and bone morphogenetic protein-7 [104]. Ebrahim et al. [105] showed that MSC-derived exosomes significantly improved diabetic nephropathy by inducing autophagy through the mechanistic target of the rapamycin (mTOR) signaling pathway in a T1D rat model.

In terms of studies **attempting to prevent T1D onset**, in a landmark study, Guay et al. retraced the pathway via which β-cell apoptosis and diabetes occurred in NOD mice [82,106,107,108]. They found that miR-142-3p, miR-142-5p, and miR155, which are generally at lower levels under normal cell homeostasis in β-cells, upon their transfer by exosomes from T lymphocytes, induced recipient cell death. Pathway analysis indicated that this transfer in turn upregulated genes involved in cytokine and chemokine signaling (Ccl2, Ccl7, CXCL10, and IFN-γ) and eosinophil-associated ribonuclease family members (Ear1, Ear2, and Ear12), which in turn was associated with a nuclear translocation of NF-kB [82]. Similar results were also observed in human islets exposed to exosomes released by activated human CD4+ T cells. Inactivation of the proapoptotic microRNAs (miRNAs) miR-142-3p, miR-142-5p, and miR-155 via the injection of a “miRNA Sponge” they engineered into four-week-old NOD mice prevented exosome-mediated apoptosis and in turn prevented T1D development [82].

Can investigation of exosomal interactions related to complications due to T1D be applied to type 2 diabetes (T2D)? Indeed, exosomes induce glucose and lipid metabolism and uptake and significantly impair gluconeogenic. Further, the cell-to-cell interactions adversely affect autophagy, thereby impairing the therapeutic restoration of homeostatic processes in T2D. In T2D, the progressive damage to the millions of β cells is adversely affected by T2D (and obesity). The interesting point is the β cell mass in obese individuals. In the T2D compensatory hyperinsulinemic, β cell destruction begins to parallel that of T1D over time. Accordingly, insights into the contribution of exosomes in improving T2D complications, including nephropathy and cardiovascular disease, may also be an area of exciting potential.

Recent investigations have revealed that exosomes induce AMP-activated protein kinase (AMPK) to stimulate autophagy, promoting glucose uptake and lipid metabolism in diabetic rats [29], and AMPK has interactions with the mammalian target of rapamycin (mTOR) signaling in autophagy regulation, as observed in other animal models of inflammation associated with cardiovascular anomalies [97]. Cardiomyocyte apoptosis and necrosis underlie cardiovascular complications in both T1D and T2D. Cardiomyocytes demonstrate a capacity in the synthesis and secretion of exosomes that promote glucose transport via mediating the levels of GLUT1 and GLUT4 to elevate glucose uptake. In turn, exosomes secreted by cardiomyocytes promote lactate dehydrogenase (LDH) activity to induce the glycolytic process of reducing glucose levels. Collectively, accumulating evidence supports a relationship between exosome and beta-cell function loss in younger-onset T1D; the degree of insulin secretion in T2D, with potential therapeutic regenerative application in childhood-onset long diabetes duration; and relationships between C-peptide levels and circulating miRNA. Further exosomes exhibit transporters involved in communication in glucose uptake and enzymes in impaired gluco- and lipid-regulatory conditions that overlap with T1 and T2D.

**Potential future applications of exosomes as nanotherapeutics.** The pathogenesis of T1D involves a complex crosstalk between insulin-secreting pancreatic β-cells and immune cells, which is partially mediated by exosomes. β-cells secrete several exosomal miRNAs that stimulate monocytes and macrophages. In turn, signals from antigen-presenting cells activate T cells, leading to the synthesis and release of several miRNAs that induce apoptosis in β-cells. This vicious cycle is terminated only after the destruction of most of the β-cell mass. As such, the miRNA component represents a potential key contributor to the dialogue between pancreatic endocrine cells and the immune system. Although the function of many of the miRNAs have not yet been elucidated, data to date suggest potential contributions to pathogenic mechanisms during the development of T1D [82,109]. Crosstalk (i.e., dialogue between pancreatic endocrine and immune cells via RNA cargo of exosomes that is also functionally transferred to β-cells) between immune cells, especially macrophages, pancreatic endocrine cells, and insulin-target tissues occurring in T1DM underlies the emerging link between exosomes and T1D. To date, the miRNA up-expression of the TLR7/8 pathway in monocytes and subsequent NK and T cell proliferation and apoptosis [109] have demonstrated binding to TLR7 in innate immune cells [110]. In addition, up-expression of miRNA in T1D has been shown to induce Ccl2, Ccl7, and Cxcl10 expression in β-cells through the translocation of NF-κB [82]. Conversely, downregulation of miRNA has been shown to inhibit β-cell apoptosis targeting CXCL10 and promote adipogenesis via HDAC9 [111]. Collectively, results demonstrate that exosomes and/or the miRNAs present contribute to β-cell dismay and the pathogenesis of T1D. However, full characterization of the role of miRNAs and other exosomal RNAs released by immune cells in β-cell survival and function is still needed.

Further, the wide range of chemical compositions of exosomes have been shown to mediate tissue regeneration in a variety of diseases, including neurological, autoimmune, and inflammatory cancer, ischemic heart disease, lung injury, and liver fibrosis. With the capacity to modulate the immune response by interacting with immune effector cells due to the presence of anti-inflammatory compounds, exosomal involvement in intercellular communication through various types of cargo highlights the potential future applications of nanotherapeutics.

## 7. Conclusions

Since the discovery of exosomes as the fourth key player in cell-to-cell signaling, much has been discovered regarding the role they play in health and in diseases like autoimmune T1D. A lot has been achieved in understanding the cell-to-cell signaling mechanisms, the immune response, and the cytokine cascade activated in T1D. Prevention of T1D has been achieved in animal models. However, several aspects will need to be addressed and multiple strategies applied before we can look to exosome engineering and RNA interference (RNAi)-based therapy to early diagnose, treat, cure, or even prevent T1D at the clinical level. RNAi therapies have the potential to ameliorate the inflammatory status of macrophages by inhibiting the secretion of inflammatory factors, downregulate the iLPS-triggered development of T1D [112], and therefore predict a pro-inflammatory state that potentially underlies the etiology. Further, depending on their cargo, the exosomes can suppress insulin resistance in T2D. Exosomes can enhance the metabolism of glucose and lipids in T2D. The enrichment of RNAi in exosomes and the underlying determination of the function of exosomes in T1D and T2D represent an auspicious area of further investigation. New technologies for the association of a specific marker with an exosome subtype and the exosome subtype with a particular function and/or group of functions warrant further investigation. In sum, exosomes represent a class of extracellular vesicles with significant roles in intercellular communication, biomolecule transport, and disease pathogenesis. Their intricate composition and versatile functions highlight the need for further scientific inquiry, holding potential for diagnostic, therapeutic, and regenerative applications across diverse fields of medicine and biology.

## Figures and Tables

**Figure 1 ijms-24-15713-f001:**
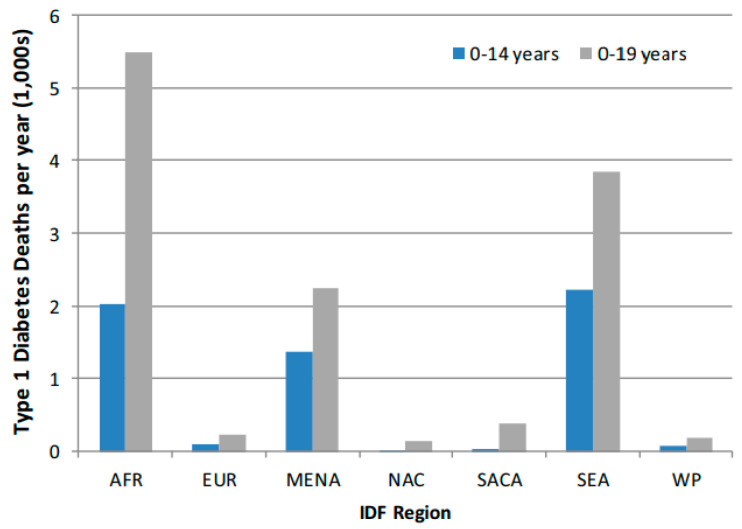
Comparison of T1D by IDF Region for 0–14 year and 0–19-year age-groups. AFR, Africa Region; EUR, Europe Region; MENA, Middle East, and North Africa Region; NAC, North America, and Caribbean Region; SACA, South and Central America Region; SEA, Southeast Asia Region; WP, Western Pacific Region. Adapted from: Patterson CC et al. [9], *Diabetes Res. Clin. Pract.* 2019; 157: 107842.

**Figure 2 ijms-24-15713-f002:**
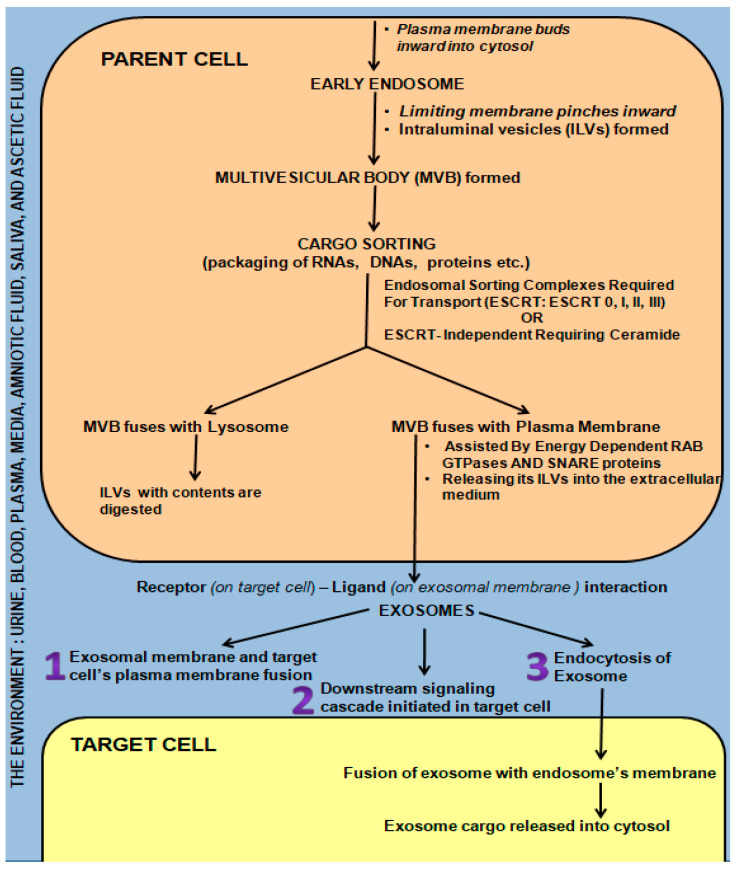
Mechanisms of Exosome Biogenesis and Secretion from “Parent cell” and Communication/Interaction with “Target Cell”.

**Figure 3 ijms-24-15713-f003:**
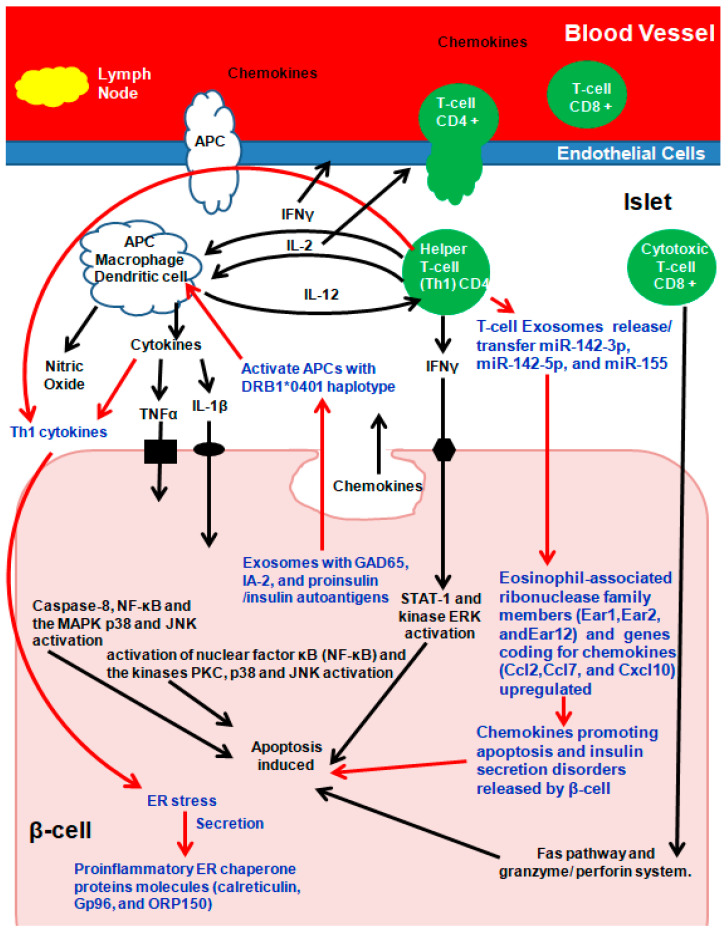
Exosomes and autoimmune Type-1 diabetes (T1D). Arrows in Red and Text in Blue present the role of Exosomes in T1D. Adapted and Modified with permission from Pirot et al., 2008 [73].

**Table 1 ijms-24-15713-t001:** The Relative Risk for the Prevalence and 4-Year Incidence of Macrovascular complications with Presence of Proliferative Diabetic Retinopathy (PDR; Adjusted for Age) in the Wisconsin Epidemiologic Study of DR.

		Relative Risk RR (95% CI)		
		Myocardial Infarction	Stroke	Amputation of Lower Extremity
**T1D**	Prevalence	3.5 (1.5–7.9)	2.6 (0.7–9.7)	7.1 (2.6–19.7)
	Incidence	4.5 (1.3–15.4)	1.6 (0.4–5.7)	6.0 (2.1–16.9)
T2D on insulin	Prevalence	0.8 (0.4–1.4)	1.2 (0.6–2.4)	4.2 (2.3–7.9)
	Incidence	1.2 (0.5–3.4)	2.9 (1.2–6.8)	3.4 (0.9–13.2)
T2D not on insulin	Prevalence	0.3 (0.0–2.4)	2.9 (0.9–9.4)	5.2 (0.6–45.0)
	Incidence	1.5 (0.2–12.5)	6.0 (1.1–32.6)	7.0 (0.8–64.4)

Adapted from: Klein R et al. [10], *Diabetes Care* 1992; 15(12): 1875–1891.

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
