# Peer review of "Exosomes; a Potential Source of Biomarkers, Therapy, and Cure for Type-1 Diabetes"

_ijms, 2023, doi:10.3390/ijms242115713_

Round 1
Reviewer 1 Report
The authors provided a well-documented overview of Ev's role and application in Type-1 Diabetes. In addition to this, the review could represent a really interesting point of view in a field so dynamic and rich in potential future applications of exosomes as nanotherapeutics. The field of research focused on exosomes is in continuous evolution, and even if the article is well written, the introduction section could be improved with a more general point of view about the application of EVs research in other fields of research adding some recent works related to the importance of exosomes in other diseases (PMID: 31936232 is just an example). A point that should be fixed is related to the size of different EV: exosomes are 30-150 nm (line 155). Figure 2 should be increased in quality and definition (objects seem to be cutted). (OPTIONAL) The conclusions could be improved and enriched by a discussion related to the need for new technologies for the association of a specific marker with an exosome subtype and the exosome subtype to a particular function and/or group of functions (PMID: 36972680 and others)
I hope that my comments could be useful for a better version of the paper.
Good luck!
Moderate editing of English language required
Author Response
Thank you for your encouraging words. We appreciate your valuable insight. We have added information in response to your comments and indeed feel your efforts have improved our manuscript
Reviewer 2 Report
The authors summarized a review on the potential roles of exosome in type-1 diabetes (T1D). The review manuscript is clinically important and will be of interest to scientists in the fields of both exosomes and T1D. However, there are some points that should be addressed/edited:
1. An inconsistency between line49-50 and line55-56. If Raposo et al. identified a function for exosomes in 1996, they couldn't have been considered "garbage" until 1998.
2. The authors alternate between using “Exo” and “exosomes” throughout the manuscript. For consistency, it would be beneficial to choose one term and stick with it.
3. Line 177-178: “exosome sub-populations that test CD9-, CD63- and CD81-positive (tetraspanins) are classified as multivesicular endosomal in origin.” It should be “extracellular vesicle (or EV) sub-populations”. exosomes are the sub-populations of EV that test CD9-, CD63- and CD81-positive (tetraspanins).
4. Incomplete sentence line 174-175: “While quantification and size estimation of exosomes is via dynamic light scattering, nanoparticle tracking analysis and surface plasmon resonance (29-31).“
5. Typos, awkward sentences, and misspellings, here are a few examples:
a. Line 238: Missing punctuation.
b. Line 428: “Ear1, Ear2, and Ear12 which was in turn was associated with a nuclear”
c. Line 433: “vented T1D development in .” Sentence does not finish.
d. Line 438: “InT2D”
e. Line 444: “Recent investigations have implicated investigation revealed that exo induced AMP-activated”
f. Line 441: “Accoridningly”
Author Response
Thank you for your in depth and comprehensive review. We found each of your comments valid, and have edited the manuscript to reflect. We apprecaite your efforts to vastly imporve our manuscript
Reviewer 3 Report
The minireview presented is interesting, but the application on type 1 diabetes is questionable
- Being an autoimmune onset on biomarker it does not seem so useful
- Even the use for a possible therapy is difficult to imagine, as the beta cells would need to be reconstituted
- miRNA could be effective in type 2 diabetes, see 10.3390/ph14121257, but in type 1 I don't see any practical applications
Therefore the work thus proposed seems incomplete to me, perhaps we could consider expanding it to type 2
It needs some revisions.
Author Response
Thank you for your valuable insight. Indeed, the breadtha nd depth of the evidence support Exo and T2D. Howeevr, emerging studies on T1D suggest therapeutic potential. We hope this is the first in a continuum of increased enthusiasm, awareness and rationale for further investigation. ALigning with comments from other reveiwers, we have added information that we think helps provide further rationale.
Round 2
Reviewer 2 Report
After evaluation, I believe the manuscript still requires major revisions. The authors have not properly addressed most of my previous comments. The inconsistent use of “Exo” and “exosomes” remains an issue. Frankly, using "Exo" as an abbreviation for a plural noun (exosomes) is confusing and inconsistent. Regarding my third comment, the revised sentence is now grammatically incorrect.
The manuscript needs major revisions to address these issues, ensuring consistency in font, format, and abbreviation usage.
The manuscript needs major revisions to ensure consistency in font, format, and abbreviation usage.
I can still detect errors.
Author Response
Thank you for re-reading our submission. Indeed, there were still several grammatical errors, which have now been corrected. We also changed all instances of "Exo" to exosome, exosomes, or exosomal (as appropriate). We are hopeful it is now in approval form.
Reviewer 3 Report
I do not see any substantial change.
it need some revision
Author Response
Thank you for re-reading our manuscript again. We added to the introduction and throughout. We feel that the review in of itself lends to the applicability to T1D. However, we added the following summary to emphasize the potentail applicability. We hope this satisfies your concerns.
The pathogenesis of T1D involves a complex crosstalk between insulin secreting pancreatic β-cells and immune cells which is partially mediated by exosomes. β-cells secrete several exosomal miRNAs that stimulate monocytes and macrophages. In turn, signals from antigen presenting cells activate T cells leading to the synthesis and release of several miRNAs that induce apoptosis in β-cells. This vicious cycle is terminated only after the destruction of most of the β-cell mass. As such, the miRNA component represents a potential key contributor to the dialogue between pancreatic endocrine cells and the immune system. Although the function of many of the miRNAs has not yet been elucidated, data to date suggests potential contributions to pathogenic mechanisms during the development of T1D (Guay et al., Tesovnik)) Crosstalk (i.e., dialogue between pancreatic endocrine and immune cells via RNA cargo of exosomes is also functionally transferred to β-cells), between immune cells, especially macrophages, pancreatic endocrine cells and insulin-target tissues occurring in T1DM underlies the emerging link between exosomes and T1D. To date, the miRNA up-expression of the TLR7/8 pathway in monocytes and subsequent NK and T cell proliferation and apoptosis as well as demonstrated binding to TLR7 in innate immune cells. In addition, up-expression of miRNA in T1D has been shown to induce Ccl2, Ccl7 and Cxcl10 expression in β-cells through the translocation of NF-κB. Conversely, down regulation of miRNA has been shown to inhibit β-cells apoptosis targeting CXCL10 and promote adipogenesis via HDAC9. Collectively, results demonstrate that exosomes and/or the miRNAs present contribute to β-cell dismay and the pathogenesis of T1D. However, full characterization of the role of miRNAs and other exosomal RNAs released by immune cells in β-cell survival and function is still needed.
Further, the wide range of chemical composition of exosomes have been shown to mediate tissue regeneration in a variety of diseases, including neurological, autoimmune and inflammatory, cancer, ischemic heart disease, lung injury, and liver fibrosis. With the capacity to modulate the immune response by interacting with immune effector cells due to the presence of anti-inflammatory compounds, exosomal involvement in intercellular communication through various types of cargo highlight the potential future applications as nanotherapeutics.
Round 3
Reviewer 2 Report
"It's 'vesicles,' not 'vessicles.'"
Other than that, I have no other comments.
Reviewer 3 Report
No others comment.